# Aspartic Aminopeptidase Is a Novel Biomarker of Aggressive Chronic Lymphocytic Leukemia

**DOI:** 10.3390/cancers12071876

**Published:** 2020-07-12

**Authors:** Pramath Kakodkar, Sanket More, Kinga András, Nikos Papakonstantinou, Sharon Kelly, Mohammad Adib Makrooni, Csaba Ortutay, Eva Szegezdi

**Affiliations:** 1Apoptosis Research Centre, School of Natural Sciences, National University of Ireland Galway, H91 W2TY Galway, Ireland; p.kakodkar1@nuigalway.ie (P.K.); more9.sanket@gmail.com (S.M.); sharon.m.kelly@nuigalway.ie (S.K.); 2HiDucator Ltd., Erämiehentie 2 E 22, 36200 Kangasala, Finland; andras.kinga89@gmail.com; 3Institute of Applied Biosciences, Centre for Research and Technology Hellas, 57001 Thessaloniki, Greece; n.papakonstantinou83@gmail.com; 4School of Mathematics, Statistics and Applied Mathematics, National University of Ireland Galway, H91 TK33 Galway, Ireland; mohammadadib.makrooni@nuigalway.ie

**Keywords:** chronic lymphocytic leukemia (CLL), DNPEP, biomarker, B-cell receptor (BcR) signaling, Bruton tyrosine kinase, phosphatidyl inositol-3 kinase, Akt, aspartic aminopeptidase, mitogen-activated protein kinase (MAPK)

## Abstract

Treatment of chronic lymphocytic leukemia has advanced substantially as our understanding of the kinase signal transduction pathways driven by the B cell receptor (BcR) has developed. Particularly, understanding the role of Bruton tyrosine kinase and phosphatidyl inositol 3 kinase delta in driving prosurvival signal transduction in chronic lymphocytic leukemia (CLL) cells and their targeting with pharmacological inhibitors (ibrutinib and idelalisib, respectively) has improved patient outcomes significantly. The kinase signaling pathway induced by the BcR is highly complex and has multiple interconnecting branches mediated by tyrosine and serine/threonine kinases activated downstream of the BcR. There is a high level of redundancy in the biological responses, with several BcR-signaling kinases driving nuclear factor kappa B activation or inducing antiapoptotic Bcl-2 genes. Accordingly, common gene targets of BcR-signaling kinases may serve as biomarkers indicating enhanced BCR-signaling and aggressive disease progression. This study used a gene expression correlation analysis of malignant B cell lines and primary CLL cells to identify genes whose expression correlated with BCR-signaling kinases overexpressed and/or overactivated in CLL, namely: AKT1, AKT2, BTK, MAPK1, MAPK3, PI3KCD and ZAP70. The analysis identified a 32-gene signature with a strong prognostic potential and DNPEP, the gene coding for aspartic aminopeptidase, as a predictor of aggressive CLL. DNPEP gene expression correlated with MAPK3, PI3KCD, and ZAP70 expression and, in the primary CLL test dataset, showed a strong prognostic potential. The inhibition of DNPEP with a pharmacological inhibitor enhanced the cytotoxic potential of idelalisib and ibrutinib, indicating a biological functionality of DNPEP in CLL. DNPEP, as an aminopeptidase, contributes to the maintenance of the free amino acid pool in CLL cells found to be an essential process for the survival of many cancer cell types, and thus, these results warrant further research into the exploitation of aminopeptidase inhibitors in the treatment of drug-resistant CLL.

## 1. Introduction

Chronic lymphocytic leukemia (CLL) is a malignancy of mature B cells. With more than 11,000 cases diagnosed every year in Europe and 15,000 in the USA, CLL is the most common type of leukemia diagnosed in the Western world [1]. CLL displays remarkable clinical heterogeneity, with many patients experiencing an indolent disease state and never requiring treatment, contrasting others who develop an aggressive disease needing treatment soon after diagnosis. Therefore, the accurate identification of patients with the aggressive disease phenotype is important.

CLL is characterized by the clonal expansion of CD5^+^ CD23^+^ B cells [2], which is driven by the accumulation of cell-intrinsic aberrations, as well as by the enhancement of B cell receptor (BcR) signaling. Approximately 55–60% of CLL patients display somatic hypermutation (SHM) in the rearranged clonotypic immunoglobulin heavy variable (IgHV) genes of the BcR (mutated CLL, M-CLL), while the remainder carry IgHV genes with minimal or no SHM (unmutated CLL, U-CLL). M-CLL is generally associated with a favorable prognosis, contrasting U-CLL, which typically exhibits an aggressive clinical course and adverse prognosis [3].

Engagement of the BcR with antigens triggers a cascade of kinase activation driving the activation of B cells. Upon activation of the BcR, the first kinases activated are the spleen tyrosine kinase (Syk) and the SRC kinase Lyn, which are recruited to the BcR complex and phosphorylate its immunoreceptor tyrosine-based activation motifs (ITAMs), namely Igα (CD79a) and Igβ (CD79b) [4]. Phosphorylation of Igα and Igβ at the ITAMs allows for the recruitment of adaptor proteins and further kinases, such as Bruton’s tyrosine kinase (BTK) and phosphatidylinositol 3-kinase (PI3K), driving the downstream activation of the prosurvival AKT pathway, followed by mTOR (mammalian target of rapamycin), NF-κB (nuclear factor kappa-light-chain-enhancer of activated B cells), and/or ERK (extracellular signal-regulated kinase) activation [4]. Studies on CLL cells demonstrated that BCR signaling is enhanced by a Zeta-associated protein of 70 kDa (ZAP70). ZAP70 is a homolog of Syk, first identified as the apical kinase in the T cell receptor-mediated signal transduction pathway. The constitutive activity of ZAP70 contributes to the untethered activation of downstream signaling kinases, leading to enhanced BCR signaling [5]. High levels of ZAP70 protein expression correlate with higher disease stages, i.e., Binet stage B or C [6]. Expression of the ZAP70 protein is also an established surrogate marker for the IgHV gene SHM status, whereby U-CLL correlates with a 5.5-fold more ZAP70 protein concentration [6,7].

Until recently, the mainstream management of CLL consisted of a combinatorial drug regimen of genotoxic agents, such as the alkylating agent cyclophosphamide and purine analogs (fludarabine) in combination with the anti-CD20 monoclonal antibody, Rituximab. However, over time, many CLL patients developed resistance against the concurrent genotoxic and cytotoxic stresses triggered by these agents. This evolution in resistance can be abstracted to the production of trophic factors released by the lymph node microenvironment and activation of BcR-signaling kinases, resulting in compensatory prosurvival signal transduction, such as induction of the antiapoptotic BCL2 [8,9] protein family.

Of the BcR-signaling kinases, the most commonly used biomarker for aggressive CLL is ZAP70. However, not all patients with a high ZAP70 expression have an aggressive disease, and, vice versa, not all patients with a low level of ZAP70 have an indolent disease. For example, in approximately 25% of patients, discordance has been observed between the ZAP70 expression and IgHV gene SHM status [10]. This observed discordance can be explained by the presence of additional genetic high-risk features, such as chromosomal deletions (e.g., 11q or 17p) [10]. Furthermore, while both 17p deletion and ZAP70 expression predict the aggressive disease, the 17p deletion falls within the ZAP70-negative patient group [10]. Moreover, there is also a high variation in ZAP70 expression depending on where the sample was collected from within the patient, i.e., peripheral blood, bone marrow, and lymph node specimen [11].

ZAP70 enhances BcR signaling by facilitating the recruitment of other kinases to the BcR complex, such as Syk, rather than via its own kinase activity [12]. Moreover, the expression of ZAP70 enhances and prolongs the activation of several key mediators of BcR signaling, such as Syk, ERK, and Akt, and decreases the rate of ligand-mediated BcR internalization. ZAP70 is also found to interact with other signaling kinases and adaptor proteins like PI3K and Shc, respectively [12]. Thus, the constitutive activation of kinases like Syk, PI3K, Akt, Btk, and ERK in response to the ZAP70 expression further drives BcR signaling, enhancing prosurvival signaling and the consequent resistance against cellular stresses triggered by chemotherapeutics. Drugs targeting key BcR-signaling kinases, particularly Btk and PI3K, have shown great success in treating patients with the aggressive disease. Especially, two oral kinase inhibitors, idelalisib (PI3K inhibitor) and ibrutinib (BTK inhibitor), have changed the standard of care for CLL patients in both the relapsed and frontline settings [13]. Despite the outstanding efficacy observed with these agents, patients still relapse, and some patients fail to respond, which is likely, since many patients have an increased activity of multiple BcR-signaling kinases, and, due to their complex interrelationship, the inhibition of only one of these kinases triggers a compensatory mechanism via other kinases and a consequent development of resistance.

In this study, we aimed to identify common or shared targets of BcR-signaling kinases as better biomarkers of aggressive CLL and potential targets to reverse drug resistance using a gene co-expression network analysis. We hypothesize that signal transduction pathways driven by the lymphoid microenvironment-activated kinases converge on common effectors. The identification of these effectors can provide novel and early biomarkers, aggressive CLL, and possible drug targets to sensitize drug-resistant CLL cells.

## 2. Results

### 2.1. Identification BCR Signaling Kinase-Correlating Genes

While ZAP70 contributes to enhanced BcR-signaling via multiple mechanisms, now it is clear that additional pathways, activated by either oncogenic transformation triggering cellular stress pathways or the lymphoid microenvironment, play an important role. Thus, in order to identify biomarkers of enhanced BcR-signaling, we identified genes whose expressions correlated with multiple BcR-signaling kinases, namely: *ZAP70*, *AKT1*, *AKT2*, *BTK*, *MAPK1*, *MAPK3*, and *PI3KCD*, using bi-weight mid-correlation analysis. For the analysis, 14 B cell gene expression microarray datasets were obtained from *ArrayExpress* at the EBI (European Bioinformatics Institute) and *Gene Expression Omnibus* (GEO) at NCBI. A description of the studies and the number of genes and samples in the datasets are summarized in Appendix A. The bi-weight mid-correlation values were first individually calculated for the 14 datasets. Then, a threshold value of 0.5 was set to select the highly correlating genes. Of these genes, there were 1262 whose expressions correlated with *ZAP70* and at least one other BcR-signaling kinase in at least five datasets (Figure 1A,B). From this list, the genes that showed correlations with multiple kinases were selected out for further analysis. The final selection contained 32 genes whose expressions correlated with ZAP70 and a minimum of two other BcR-signaling kinases (Appendix A). Of these 32 genes, the ones that correlated with *ZAP70* and *AKT1/2* expressions also correlated with *MAPK3* and *PI3KCD* expressions but not with *BTK* (Figure 1C,D). Interestingly, there was a relatively small overlap between *AKT1/2* and *PI3KCD* co-expressed genes. Many of the genes that correlated with *ZAP70* and *PI3KCD* also showed co-expressions with *MAPK3* but not with *AKT1/2* or *BTK*, indicating that the gene expressional programs associated with different BcR-signaling kinases are diverse and only partially overlapping (Figure 1C,D).

A network analysis found that 28 of the 32 genes formed a closely connected, minimal network, clustering around four main nodes: HNF4A (hepatocyte nuclear factor 4 alpha), EED (embryonic ectoderm development), ELAVL1 (ELAV-like RNA binding protein 1), and MAPK1/3 and that the 32-gene signature reports on the activity of these four genes/pathways. This well-interlinked signaling network (Figure 1E) contains nodes already known to have a role in CLL, such as EZH2 and NF-κB, and also identified new pathways not well-associated with CLL (HNF4A and ELAVL1 nodes) [14,15,16].

### 2.2. DNPEP Is a Prognostic Marker of Aggressive CLL

Further analysis was directed towards validating the prognostic power of the identified genes by analyzing the time to treatment and overall survival responses using an independent transcriptomic dataset of 107 CLL patients [17]. For both analyses, the hazard ratio associated with ZAP70 mRNA expression was used as a baseline for comparison. Regarding time-to-treatment, a high *ZAP70* mRNA expression was associated with a hazard ratio of 1.45 (of note, the clinically used Zap70 expression measure, the percentage of Zap70 protein-expressing cells, was not recorded in the dataset; thus, we used the mRNA expression values). As a measure of their prognostic potential, the HR values associated with the time to treatment for the 32 genes were determined individually (Figure 2A), as well as together (Figure 2B). When analyzed together, the 32-gene set could clearly segregate low and high-risk groups with an HR value of 24.49 (Figure 2B). When the 32 genes were analyzed individually, 8 out of the 32 genes (*DNPEP, ARPC4, SMARCA4, TUBA1C, PSMD13, VPS39, EBI3,* and *ESYT1*) were associated with an HR higher that of ZAP70 (Figure 2A), and combined, these eight genes could segregate the patients into high and low-risk groups with an HR of 2.99 (Figure 2C). Additionally, the contribution of each gene to the combined HR was assessed by the iterative removal of each gene, one at a time, from the 32-gene pool (Appendix A). The strongest eight contributors to the HR were *PCSK7, CYC1, VPS72, PSMD13, RPL38, SLC25A11, DNPEP*, and *ISOC2*. This eight-gene signature was associated with an HR value for a time to treatment of 3.47 (Appendix A).

The same analysis as above was carried out for the overall survival. The reference *ZAP70* expression was associated with an HR of 1.94, while the hazard ratio associated with the combination of the 32 BCR kinase-correlating genes was 32.93 (Figure 3B). An analysis of the 32 genes individually (Figure 3A) revealed eight genes associated with a high HR, and the combination of these eight genes could segregate the patients into high and low-risk groups, with an HR of 6.87 (Figure 3C). This cluster of eight genes consisted of *PSMD13, DNPEP, TUBA1C, RPL38, ISOC2, ESYT1, SMARCA4,* and *ST6GAL1*. The contribution of each gene to the combined HR was also assessed by the iterative removal of each gene, one at a time, from the 32-gene pool (Appendix A). The strongest eight contributors of this analysis included *ARPC4, EBI3, RPP7A, PCSK7, DNPEP, Corf61, PDXK,* and *CWF19L1,* and, combined, these eight genes could segregate the CLL patients into high and low-risk groups with an HR of 7.47 (Appendix A).

Overall, from the analyses at the individual gene level, four genes were found to correlate with both the time-to-treatment and overall survival: *DNPEP, PSMD13, ESYT1*, and *SMARCA4*. From the reciprocal analysis (iteratively removing one gene out of the pool of 32), only *DNPEP* was common between the time-to-treatment and overall survival, and *DNPEP* emerged as the common gene across the four analyses.

### 2.3. DNPEP as a Prognostic Marker for CLL

By dichotomizing the patients from the Herold dataset into high- and low-*DNPEP* expressing groups based on the median *DNPEP* expression, patients with high *DNPEP* expressions tended to have a shorter time to treatment, as well as overall survival probability (Figure 4A,B). To further validate *DNPEP* as a marker for prognosis, *DNPEP* transcript levels were measured in 34 primary CLL patient samples with real-time quantitative RT-PCR, and a correlation with Rai and Binet staging was determined. The clinical information of the samples is summarized in Appendix A. The *DNPEP* mRNA expression correlated with Rai staging, with the *DNPEP* expression of the asymptomatic stage (0) being significantly lower than a high risk (III-IV) (*p* = 0.0019). Similarly, *DNPEP* transcript levels of medium-risk (I-II) and high-risk (III-IV) groups were also significantly different (*p* = 0.0014, Figure 4C). The *DNPEP* expression also correlated with Binet stages, with a high *DNPEP* expression detected in the intermediate and high-risk groups (B and C stages, *p* = 0.024, Figure 4D).

To determine whether *DNPEP* is an independent prognostic marker, it was compared to the immunoglobulin variable heavy chain (IGVH) gene mutational status and 17p13 deletion in a multivariate (Cox) analysis looking at the time to treatment and overall survival using the Herold dataset [17]. In the database, 48% of the patients had mutated IGVH, 45% had unmutated IGVH, and 7% was unknown, while 8% of the patients carried 17p13 deletion. A high *DNPEP* expression (based on the median expression) was a significant prognostic factor in the univariate analysis, both for the time to treatment and overall survival (*p* = 0.009 and *p* = 0.01, respectively). In the multivariate Cox regression analysis, a high *DNPEP* expression remained prognostic for the time to treatment after adjusting for the IGVH mutational status and 17p13 deletion but not for the overall survival (*p* = 0.032 and *p* = 0.224, respectively).

### 2.4. Inhibition of DNPEP Increases Sensitivity to PI3K- and BTK-Inhibitions

In order to investigate whether *DNPEP* is only a biomarker or a possible therapeutic target, a recently developed inhibitor against *DNPEP* was used [18,19]. The *DNPEP* gene codes for an N-terminal peptidase selective for aspartic acids. The inhibitor, DI93293, used in our study could target two aminopeptidases, DNPEP and the closely related ENPEP (glutamine aminopeptidase). The inhibitor inhibited the peptidase activity of recombinant DNPEP by 95.0% ± 1.4% [18] and by 88% in Mec-1 CLL cell lysate (at 12.5 μM and 25 μM, respectively, Appendix A). The IC_50_ value of the inhibitor was 1.4 μM [18].

To test the effect of the *DNPEP* inhibition on CLL cell survival and drug resistance, Mec-1 CLL cells were cultured alone or on a mesenchymal stromal cell layer expressing the CD40 ligand (CD40L) as a model of the CLL microenvironment. After 24 h of culture, the cells were exposed to a dosage of DI93293 alone or with the Bcl-2/Bcl-X_L_ inhibitor, ABT737, for 48 h, and the induction of cell deaths in Mec-1 cells were determined with annexin V staining (after the exclusion of stromal cells based on their green fluorescent labels). The inhibition of *DNPEP* alone did not induce any cell deaths but had a modest potentiating effect on ABT737-induced cytotoxicity shown by the CI indices of lower than 1 for all drug dose combinations (calculated by the Chou-Talalay method; Figure 5A, tables under the plots). Stromal support of the CLL cells mostly affected the ABT737 sensitivity, but the potentiating effect of DI93293 was retained (Figure 5B).

The effect of *DNPEP* inhibition on the drug sensitivity of primary, patient-derived CLL cells was also tested using ABT737 and two kinase inhibitors, the PI3K inhibitor, idelalisib, and the BTK inhibitor, ibrutinib (Figure 5C–E). Mononuclear cells isolated from five patients were cultured with bone marrow mesenchymal stromal cell support for 24 h before exposing the cultures to DI93293 alone or in combination with the above drugs for 48 h. The induction of cell death was determined with To-Pro3 staining in the CLL cell population. We found that the inhibition of *DNPEP* potentiated cell deaths induced by both idelalisib and ibrutinib but not ABT737 (Figure 5D,E).

## 3. Discussion

The identification of biomarkers able to indicate the activity of several BcR-linked kinases would improve patient stratification. By using a bi-weight gene expression correlation analysis, we identified genes whose expressions correlated with several BCR-signaling kinases. Genes whose expression could predict patient prognosis correlated with *MAPK1* and/or *MAPK3*, *PI3KCD*, and *ZAP70* and *PI3KCD*, indicating that the gene expressional program’s driving expressions of BcR-signaling kinases are diverse and only partially overlapping.

By selecting the genes whose expressions correlated with ZAP70 plus two other BCR-signaling kinase gene, we identified a 32-gene panel. The 32-gene panel could prognose the time to treatment and overall survival probability with very high HR values of 24.49 and 32.93, respectively. The reduction of the number of genes by selecting the top quartile from each analysis, while still producing a prognostic gene set, was nearly not as robust as the 32 genes together, highlighting again that the BCR-induced kinase signaling pathways are only partially overlapping. One gene, aspartic aminopeptidase (*DNPEP*), showed a high HR value and consistently appeared in all analyses. This biomarker can predict and stratify the risk when CLL patients develop the progressive disease, requiring therapy. An analysis of the *DNPEP* expression in combination with the IGV_H_ mutation status could be used as a prognostic tool to detect when CLL patients develop the progressive disease and, therefore, when treatment would be required.

*DNPEP*, a member of the M18 peptidase family, is a zinc/manganese-containing metallopeptidase expressed in the cytosol [13]. *DNPEP* cleaves N-terminal aspartate residues from proteins and peptides. Aminopeptidases execute the final step of intracellular protein degradation by trimming or fully degrading peptides produced by the ubiquitin-proteasome pathway. Partial peptide degradation by aminopeptidases may be used to generate MHC-presented peptide antigens, or, after full hydrolysis, the generated free amino acids can be reutilized by the CLL cells for protein synthesis.

Although, to date, little is known about the physiological functions of *DNPEP*, the enhanced expressions of other aminopeptidases in cancer are long known. For example, pancreatic cancer, lymphoma, and leukemia patients have increased leucine aminopeptidase (LAP) activity. Aminopeptidase N (APN) is another type of aminopeptidase implicated in human cancers, such as thyroid cancer [20], ovarian carcinoma [21], breast [22,23], and colon cancer [24]. Moreover, APN expression was shown to correlate with poor survival, as well as decreased disease-free survival in colon cancer [24].

Further indicating the potential role of aminopeptidases in cancer, the aminopeptidase inhibitor [25] bestatin showed efficacy in lung cancer [26]. New aminopeptidase inhibitors are also emerging, with the best-known example of the prodrug, tosedostat, which is currently in phase II clinical trials for acute myeloid leukemia [25,27].

These results show that aminopeptidases are not only biomarkers of various cancers, but they have a biological functionality in cancer. The free amino acids produced by aminopeptidases support protein synthesis, which some cancers strongly rely on. Accordingly, the inhibition of aminopeptidases by CHR79888 (the active metabolite of tosedostat) induced a typical amino acid deprivation response (AADR) in HL-60 AML cells, including the inhibition of mTOR phosphorylation, reducing the protein synthesis and accumulation of intracellular peptides [27]. New emerging results, which show that leukemic cells are sensitive to amino acid deprivation, further support the rationale of aminopeptidase inhibitors as cancer therapeutics [28].

## 4. Materials and Methods

### 4.1. B Cell Transcriptomic Datasets and Preprocessing

B cell transcriptomic datasets were obtained from two main public databases, the *ArrayExpress* at the EBI (European Bioinformatics Institute, https://www.ebi.ac.uk/arrayexpress/) and the *Gene Expression Omnibus* (GEO) at NCBI (National Centre for Biotechnology Information, https://www.ncbi.nlm.nih.gov/geo/). Fourteen B cell transcriptomic datasets were selected. Preprocessing of gene expression datasets included normalization and annotation of the datasets using the open-source programming language “R”. Datasets from single-channel microarray datasets were normalized using the Robust Multi-Array Average (RMA) method, while two-channel microarray datasets were normalized using the locally weighted scatterplot smoothing (LOESS) method. These normalized datasets were annotated by using BiomaRt and Bioconductor packages.

### 4.2. Weighted Gene Co-Expression Network Analysis

For calculating correlation values between BCR-signaling kinase genes and genes in microarray datasets, the WGCNA package [29] was used, which implements the necessary functions for correlation calculation, gene selection, cluster identification, and network construction. Bi-weight mid-correlation was used to identify genes whose expression correlates with BCR-signaling kinases. The gene list obtained was filtered by applying an absolute threshold and ranking the genes in descending order by the number of datasets the gene was present in for each kinase.

### 4.3. Statistical Analyses

For time-to-treatment and overall survival analyses, an independent transcriptomic dataset from 107 CLL samples analyzed on Affymetrix HG-U133 Plus 2.0 chips was used [17,30]. For analyses where the combined effect of several genes was assessed, the sample set was divided into two equal-sized risk groups by ranking and splitting the samples at the median according to their estimated prognostic index (risk score). The prognostic index was calculated based on the beta coefficients multiplied by gene expression values using Cox proportional hazard regression: (h*_i_*(t)/h*_0_*(t) = exp(β*_1_x_1_*+ β*_2_x_2_+ …* β*_i_x_i_*), h*_i_* is the hazard of the i-th individual, h*_0_* is the baseline hazard function, exp is the exponent function, *x_i_* is the expression value and the *β_i_* can obtained from the Cox fitting and the term within the exp function is the prognostic index or risk score. In case a single gene was analyzed, the dataset was divided into two groups based on the median expression level of the gene of interest. Cox proportional hazards models to model the probability of survival and the time to treatment were used where the DNPEP expression (as a binary variable indicating low and high-risk individuals based on the median DNPEP expression) was the only predictor. For multivariate analysis, Cox proportional hazards models were used to model the probability of the survival and treatment with three predictors: IGV_H_ mutational status, p17 mutational status, and DNPEP expression. The Coxph function in the Survival package in R was used to compute the Cox Proportional Hazard regression models and obtain the hazard ratios. The dependent variable, which is a survival object in this linear model, was created using the Surv function in Survival package. Additionally, the Survminer package in R was used to plot the survival curves. The dynamic nomogram of these models, along with the underlying model summaries, have been provided by the DynNom package [31] in R: https://adibmakrooni.shinyapps.io/cxmod10/, https://adibmakrooni.shinyapps.io/cxmod4/, https://adibmakrooni.shinyapps.io/cxmod1/, and https://adibmakrooni.shinyapps.io/cxmod2/. The expression values of DNPEP in the primary CLL samples were analyzed in the patients grouped by the Rai and Binet clinical CLL stages. Regarding the Rai stages, stage 0 patients (*n* = 20) were handled as low risk, stage I-II patients (*n* = 10) as intermediate, and stage III-IV patients (*n* = 4) as high risk. With Binet stages, only stage A patients (*n* = 29) and stage B and C patients (*n* = 5) were distinguished. To compare the mean expression levels of the groups, a Welch two-sample *t*-test was used in the R statistical environment.

### 4.4. Primary CLL RNA Samples

RNA isolated from peripheral blood mononuclear cells of 34 CLL patients were used for gene expression analyses. All patients consented, and the study had ethical approval, research ethics committee: NUI Galway, permission date: 01-06-2015 (duration: 60 months); reference number: CA01355. A summary of the patient clinical data is provided in Appendix A.

### 4.5. Reverse Transcription-Coupled Quantitative Real-Time Polymerase Chain Reaction

Reverse transcription of patient RNA samples was performed using 1 µg of RNA sample with Superscript II (Invitrogen, Waltham, MA, USA), according to the manufacturer’s instructions. The expression of genes was analyzed by using PrimeTime Predesigned qPCR assays (Integrated DNA Technologies, IDT, Coralville, IA, USA). Abelson murine leukemia viral oncogene homolog 1 (*ABL1*) was used as a housekeeping gene for normalization. The TaqMan primer and probe sets for all genes (*ABL1, ZAP70*, and *DNPEP*) were predesigned sets purchased from Integrated DNA Technology. For the quantitative polymerase chain reaction (qPCR), Agilent Brilliant III qPCR master mix was used. Each qPCR was performed in a 6-µL reaction volume containing 0.3 µL of cDNA, 2.1-µL H_2_O, 3.0 µL of qPCR master mix, and 0.6 µL of the qPCR primer assay. Initial denaturation at 95 °C for 15 min was followed by 40 cycles of a denaturation step at 95 °C for 15 s, an annealing step at 57.5 °C for 30 s, and an extension step at 72 °C for 30 s on a Roche LC480 qPCR light cycler.

The threshold cycle (Ct) value was calculated as the cycle number at which the fluorescence of the reporter reached a fixed threshold. The calculation of the relative expression of the target genes in comparison to the reference gene was performed in an R statistical environment (Appendix A).

### 4.6. Cell Culture

The Mec-1 CLL cell line was cultured in RPMI-1640 medium (Sigma, St. Louis, MO, USA) containing 10% heat-inactivated FBS (Hyclone, Marlborough, MA, USA), penicillin (100 U/mL), streptomycin (100 μg/mL) (Sigma), and L-glutamine (2 mM). The bone marrow mesenchymal stromal cell line, HS-5, was cultured in DMEM (high-glucose, Gibco, Waltham, MA, USA) supplemented with 10% FBS (Hyclone), GlutaMAX-I (2 mM) (Gibco), penicillin (100 U/mL), streptomycin (100 μg/mL), and 1-mM sodium pyruvate (Sigma). Primary CLL samples were generated from peripheral blood samples by isolating the mononuclear cell (MNC) fraction with Ficoll gradient-centrifugation [32]. MNCs were stored in liquid nitrogen until use. Upon revival of the cells, viability was determined with trypan blue staining. Only samples with viability above 60% were used. All patients consented according to the local Ethical Committee Regulations. Primary CLL cells were grown on an HS-5 stromal feeder layer in RPMI-1640 medium (Sigma) containing 10% FBS, penicillin (100 U/mL), streptomycin (100 μg/mL), and L-glutamine (2 mM). The HS-5 cells were stably transfected with GFP (green fluorescent protein) to enable their identification and separation from the CLL cell population in downstream analyses.

### 4.7. Measurement of Cell Viability

Mec-1 cell viability was quantified with annexin V staining. Cells were collected and stained with annexin V-APC (Immunotools) in annexin V buffer (10-mM HEPES/NaOH, pH 7.5, 140-mM NaCl, and 2.5-mM CaCl_2_) for 15 min on ice in the dark. Samples were analyzed on a FACS Canto II flow cytometer. The viability of primary CLL cells was determined using the viability dye, To-Pro-3 (Molecular Probes), according to the manufacturer’s protocols. The samples were analyzed using the BD FACS Canto II flow cytometer (BD Bioscience, San Diego, CA, USA) by collecting 30,000 events. HS-5 cells were excluded from the analysis by gating out GFP^+^/FSC^high^ events. Statistical analysis was performed using FCSExpress (DeNovo Software Inc.,Pasadena CA, USA) and GraphPad Prism (GraphPad Software Inc., La Jolla, San Diego, CA, USA) software packages.

### 4.8. DNPEP Enzyme Activity Assay

L-aspartic acid 7-amido-4-methylcoumarin (Asp-AMC) was used as the substrate to analyze the DNPEP enzyme activity. Mec-1 cell lysate, as the source of DNPEP enzyme, was obtained by lysing 1 × 10^6^ Mec-1 cells in 100 uL of cell lysis buffer (250-μM HEPES, pH 7.4, 25-μM CHAPS, and 2-μM DTT). Ten microliters of cell lysate was dispensed into a black, clear-bottom 96-well plate and incubated with 100-μL assay buffer (50-mM Tris-HCl, pH 7.5) containing 22-μM ASP-AMC substrate at 37 °C; As a negative control, DNPEP activity was blocked by adding 0.5-mM ZnCl_2_ to the reaction. DNPEP activity was monitored kinetically by determining the fluorescence produced in every 60 s over 30 min. Enzyme activity was calculated from the linear portion of the fluorescent intensity graphs. The effect of the DNPEP inhibitor (DI93293) was calculated according to the following formula: activity score = ((RFI_DI_ − RFI_u_)/(RFI_u_ − RFI_Zn_)) × 100%, where RFI_u_: enzyme activity of untreated sample, RFI_DI_: enzyme activity of DI-treated sample, and RFI_Zn_: same of sample containing ZnCl_2_.

## 5. Conclusions

In conclusion, by using gene co-expression analysis, we identified DNPEP as a biomarker to identify CLL patients with poor prognosis. This biomarker can predict and stratify the risk of when CLL patients develop the progressive disease, requiring therapy. This is comparable to the Rai and Binet clinical CLL staging systems. DNPEP, in combination with IgHV, could be used as a prognostic tool for when CLL patients develop the progressive disease, requiring therapy. Furthermore, our results warrant further study evaluations of DNPEP as a novel target for combination therapy to reverse drug resistance.

## Figures and Tables

**Figure 1 cancers-12-01876-f001:**
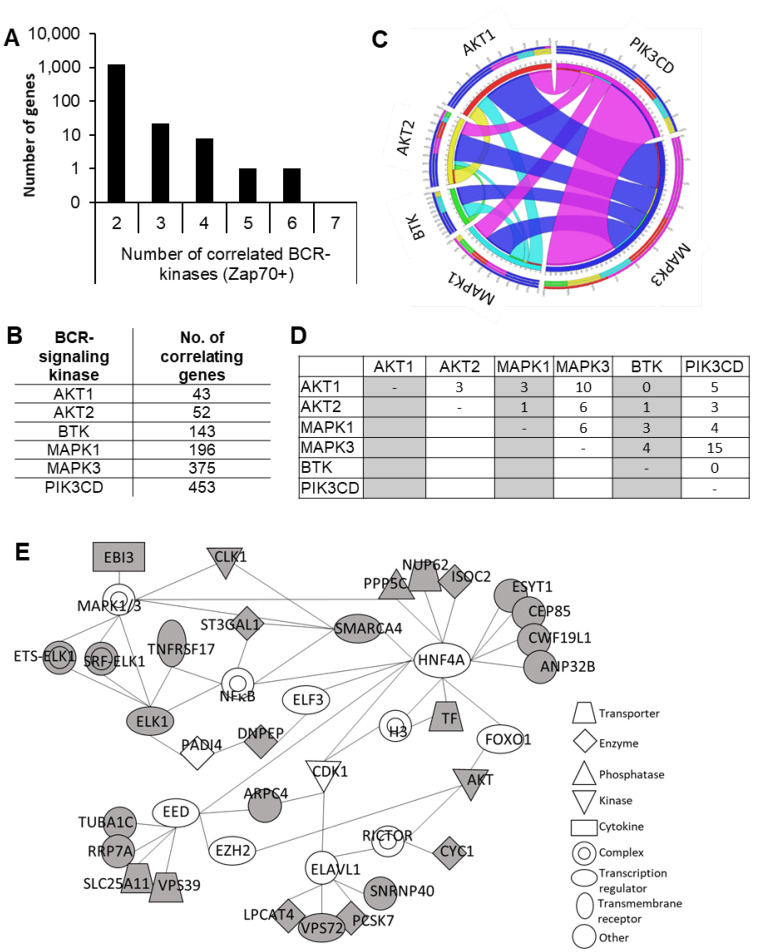
Partial integration and diversity in gene expressional programs that drive the expressions of BCR-signaling kinases. (**A**) Total number of genes whose expressions correlated with that of *ZAP70* and at least one other BCR-signaling kinase in at least five datasets. (**B**) The number of correlating genes identified for each BCR-signaling kinase. (**C**) Circos plot showing the distribution of common targets of BCR-signaling kinase pairs. (**D**) Matrix representation of the number of genes that are common correlating genes of BCR-signaling kinase pairs. (**E**) Interaction network of the 32 genes identified. Ingenuity pathway analysis was carried out to identify gene networks the 32 BCR-signaling kinase co-expressed genes reported on. Grey-shaded genes are the identified BCR-kinase correlating genes.

**Figure 2 cancers-12-01876-f002:**
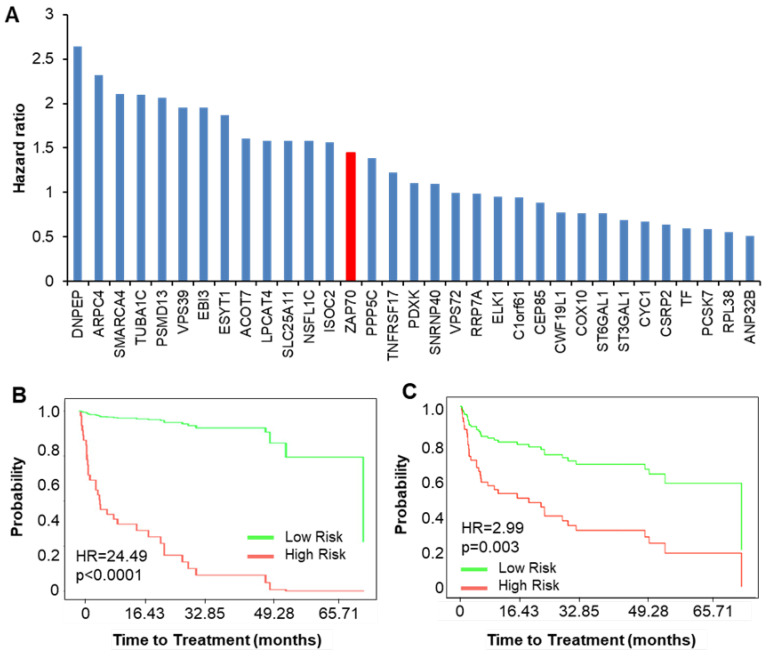
Prognostic potential of genes correlating with BCR-signaling kinases regarding the time to treatment. (**A**) Bar graph showing hazard ratios associated with each of the 32 genes in the panel. The hazard ratio (HR) of ZAP70 (red bar) is added as a reference. (**B**,**C**) Kaplan Meier plots showing time-to-treatment probability curves for the 32 genes together (**B**) and the combination of the eight genes with the highest HR values (from graph in part A, (CI: 1.44 to 6.21))**.** The *y*-axis represents the estimate probabilities, and the *x*-axis represents the time to treatment (months).

**Figure 3 cancers-12-01876-f003:**
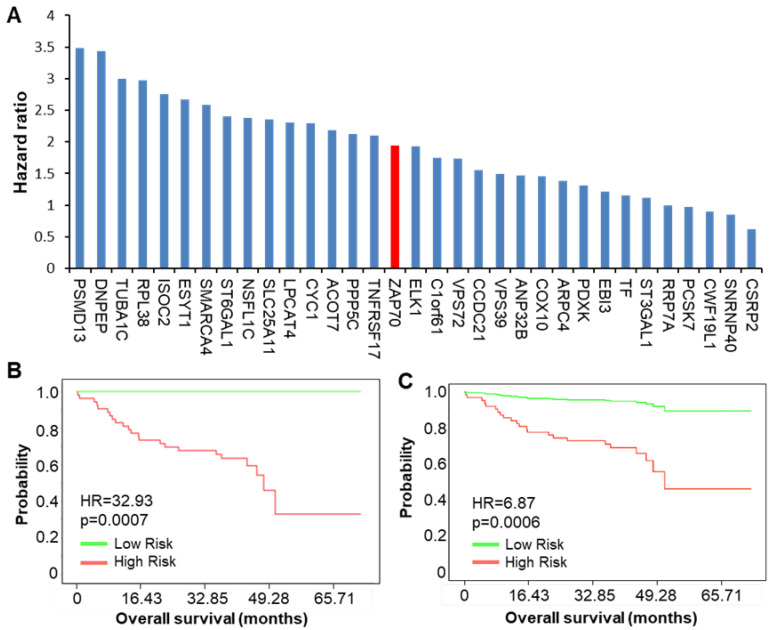
Prognostic potential of genes correlating with BCR-signaling kinases regarding the overall survival. (**A**) Bar graph showing the hazard ratios for the overall survival associated with each of the 32 genes in the panel. The HR of ZAP70 (red bar) is added as a reference. (**B**,**C**) Kaplan Meier plots showing the overall survival probability curves for the 32 genes together (CI: 4.4 to 246.3) (**B**) and the combination of the eight genes with the highest HR value (from graph in part A, (CI: 2.28 to 20.67, *p* = 0.0006)). The *y*-axis represents the estimate probabilities, and the *x*-axis represents the time to treatment (months).

**Figure 4 cancers-12-01876-f004:**
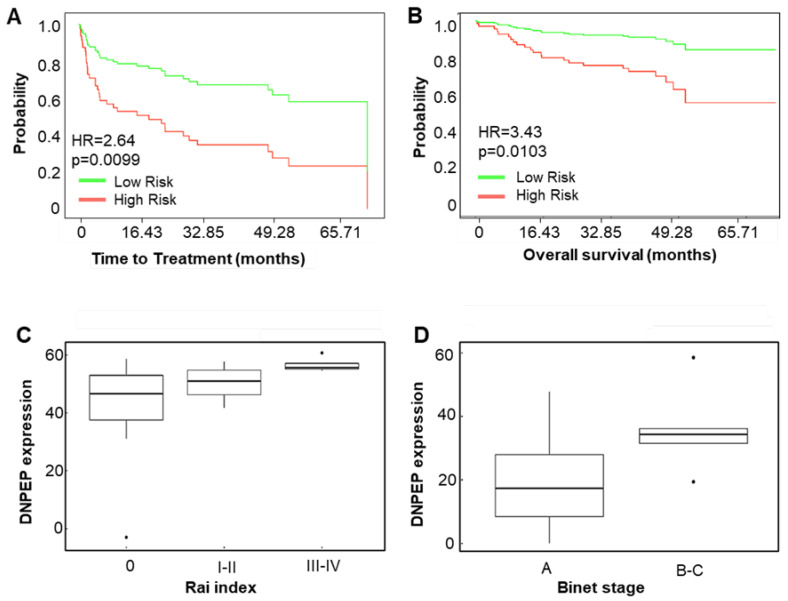
Prognostic potential of *DNPEP*. Kaplan-Meier plots for time-to-treatment (**A**) and overall survival (**B**) for *DNPEP* expression (CI: 1.26 to 5.54 and CI: 1.34 to 8.79 for A and B, respectively). (**C**,**D**) Correlation between *DNPEP* expression and disease stage. Patient samples were divided into low risk and high risk based on the median *DNPEP* mRNA expression, and the time to treatment and overall survival probabilities were graphed. The *DNPEP* mRNA expression was determined by quantitative real-time (RT)-PCR from 34 chronic lymphocytic leukemia (CLL) patient RNA samples. The box plots show the *DNPEP* expressions by Rai (**C**) and Binet (**D**) staging. (**C**) The difference between the *DNPEP* transcript levels of the Rai medium-risk (I–II) and high-risk (III-IV) groups and between the asymptomatic stage (Rai 0) and high risk (III–IV), as well as (**D**) between the early stage (Binet A) and late metastatic stages (Binet B and C), were significant (*p* = 0.0019, *p* = 0.0014, and *p* = 0.024, respectively).

**Figure 5 cancers-12-01876-f005:**
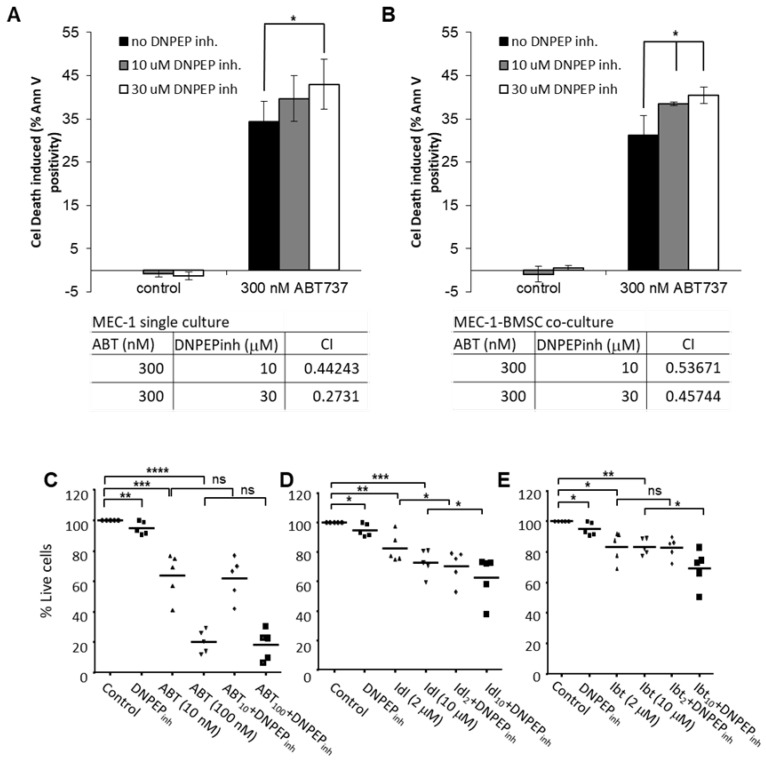
(**A**) The inhibition of *DNPEP* activity potentiates AB737-induced CLL cell death. Mec-1 CLL cells were treated with 300 nM of the BH3-mimetic (ABT737) and with the indicated doses of the developmental *DNPEP* inhibitor DI93293 for 48 h—after which, the percentage of dead cells was determined with the annexin V assay. * *p* < 0.05. (**B**) The potentiating effect of *DNPEP* inhibition is retained in a microenvironment-modeling culture of CLL. Mec-1 cells were seeded on a monolayer of CD40L-expressing bone marrow stromal cells for 24 h before exposing them to 300-nM ABT737 and indicated doses of the *DNPEP* inhibitor DI93293 for 48 h. Cell deaths induced were determined with the annexin V assay. The tables under the graphs of parts (**A**,**B**) list the combination indices calculated using the Chou-Talalay method (Compusyn). (**C**–**E**) The potentiating effect with *DNPEP* inhibition on primary CLL cells was tested on ABT737 (ABT), the PI3K inhibitor (idelalisib. Idl), and the BTK inhibitor (ibrutinib, Ibt). Mononuclear cells isolated from 5 patients were cultured with bone marrow mesenchymal stromal cell support for 24 h before exposing the cultures to DI93293 alone or in combination with the above drugs for 48 h. The induction of cell death was measured with To-Pro3 staining in the CLL cell population. *p*-values: * *p* < 0.05, ** *p* < 0.01, *** *p* < 0.001, and **** *p* < 0.0001.

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
