# Peer review of "Aspartic Aminopeptidase Is a Novel Biomarker of Aggressive Chronic Lymphocytic Leukemia"

_cancers, 2020, doi:10.3390/cancers12071876_

Round 1

Reviewer 1 Report

All my concerns have been addressed. Please updated references list as you have added more in the introduction.

Author Response

We appreciate the reviewer's comments and we have carried out revisions of our manuscript to address the reviewers’’ comments. We give a point by point response to the points the reviewers raised below and the changes in the manuscript are highlighted as red coloured text.

Reviewer 1

Comment: All my concerns have been addressed. Please updated references list as you have added more in the introduction.

Response: We have checked all references cited in the introduction and confirmed that they are all present in the reference list.

Reviewer 2 Report

This study demonstrated that DNPEP expression is an independent prognostic factor for time-to-treatment. Moreover, the study also highlights the interest of DNEPP as a therapeutic target.

Many critical points that have been considered in the previous revision and have been addressed adequately by the Authors. As many aspects of this study have already been discussed, my review is focused mainly on the discussion of the results.

I would suggest changing:

 “CLL may change from an indolent to an aggressive…."  and "CLL may become aggressive" with "when patients develop progressive disease requiring therapy."

The sample size of the analyzed patients is small. Please mention how the 34 patients included in this study were selected. Information about some variables included in the IPI prognostic score is lacking. It is unclear if patients were or not previously treated.                                                                                                       The DNPEP expression could be associated with other prognostic factors, as well as previous treatment.

Multiple and complex statistical analyses are described in the Result section. Due to the multitude of data, results are not very easy to follow. Whenever possible, I would suggest describing some details in the supplementary materials.

The independent prognostic value of DNEPP is a relevant finding. Do the authors believe that DNEPP assessment may improve the prognostic stratification of CLL patients?  Did you observe inferior survival in cases with both, DNEPP overexpression and unmutated IGHV?

The results of this work suggest that DNPEP peptidase can be considered as a potential therapeutic target. This finding should deserve more space in the discussion.

.....pancreatic cancer, lymphoma and, leukemia patients have increased leucine aminopeptidase (LAP) activity ..

Please introduce more details on this point and its reference.

Have you evaluated whether the DNEPP expression varies in the same patient in relation to the response to ibrutinib therapy?

Author Response

We appreciate the reviewer's comments and we have carried out revisions of our manuscript to address the reviewers’’ comments. We give a point by point response to the points the reviewers raised below and the changes in the manuscript are highlighted as red coloured text.

Comment: I would suggest changing: “CLL may change from an indolent to an aggressive…."  and "CLL may become aggressive" with "when patients develop progressive disease requiring therapy."

 Response: We agree with the reviewer and have complied with the following changes:

  • Line 288-291: Revised as requested to “when CLL patients develop progressive disease” instead of “when CLL may become aggressive disease”
  • Line 421: Revised as requested to “when patients develop progressive disease requiring therapy” instead of “CLL may change from indolent to an aggressive disease”
  • Line 424: Revise as requested to “when patients develop progressive disease requiring therapy” instead of “when CLL becomes aggressive”

Comment: The sample size of the analyzed patients is small. Please mention how the 34 patients included in this study were selected. Information about some variables included in the IPI prognostic score is lacking. It is unclear if patients were or not previously treated. The DNPEP expression could be associated with other prognostic factors, as well as previous treatment.

Response: Patient samples collected at diagnosis were used without pre-selecting for any clinical features. Unfortunately, we do not have information on serum b2 microglobulin concentration, thus it is not included in the clinical characteristics table. We agree that DNPEP expression could be correlated with other prognostic factors, but given the small sample size, we did could not pursue it in the currently available datasets. It will be addressed in future, follow-up studies.

Comment: Multiple and complex statistical analyses are described in the Result section. Due to the multitude of data, results are not very easy to follow. Whenever possible, I would suggest describing some details in the supplementary materials

Response: We appreciate the reviewer’s comment and thus we reviewed the details we provide in the manuscript (Line: 348-362) for the statistical analyses carried out including the biweight midcorrelation analysis of the gene expression data, hazard ratio determination, univariate and multivariate correlation analyses. We have also provided the links for the analysis tool and our setup used for the multivariate analyses. We hope that these additions provide clarification on the methods used.

Comment: The independent prognostic value of DNPEP is a relevant finding. Do the authors believe that DNPEP assessment may improve the prognostic stratification of CLL patients?

 Response: We agree with the reviewer that DNPEP may be a valuable prognostic marker and our results warrant further studies to explore this question in larger patient datasets.

 Comment: Did you observe inferior survival in cases with both, DNEPP overexpression and unmutated IGHV?

Response: Our data showed stronger prognostic value for time-to-treatment. In multivariate Cox regression analysis, high DNPEP expression remained prognostic for time to treatment after adjusting for IGVH mutational status and 17p13 deletion, but not for overall survival (p=0.032 and p=0.224, respectively).

Comment: The results of this work suggest that DNPEP peptidase can be considered as a potential therapeutic target. This finding should deserve more space in the discussion. Pancreatic cancer, lymphoma and, leukemia patients have increased leucine aminopeptidase (LAP) activity. Please introduce more details on this point and its reference. Add a 1-2 additional examples of other aminopeptidease, which have been shown to highly expressed in some tumours.

Response: As the reviewer suggested we have now added examples of other aminopeptidases known to be overexpressed in cancers and their prognostic value (Line: 302-308).

Comment: Have you evaluated whether the DNPEP expression varies in the same patient in relation to the response to ibrutinib therapy?

Response: We have only evaluated the effect of DNPEPinh with Ibrutinib on CLL cell survival and we have not evaluated whether DNPEP expression would change in response to ibrutinib. While DNPEP expression did not show correlation with BTK expression in our transcriptomic analysis, it does not exclude that DNPEP expression correlates with BTK activity.

This manuscript is a resubmission of an earlier submission. The following is a list of the peer review reports and author responses from that submission.

Round 1

Reviewer 1 Report

This is an interesting and well undertaken study.

 I have a few minor comments

Figure 1A:  the figure axis could be modified to indicate that if it is ZAP70 plus for example 2 others or if it includes ZAP70 as part of the "2"?

How was the high and low ZAP70 groups defined - was it median expression?

Figure 2B - how can ZAP70 be part of the group of genes correlating with ZAP70 and minimum of two other BCR-signalling kinases - or was the red ZAP70 bar only present to show the individual hazard ratio?

Figure 4A and B - no p values are shown to support the statement of "small but significant" effect.  Have the authors calculated combination index scores such Chou-Talalay scores?

Did the authors undertake any network or pathway analysis on the 32 genes?

Have the authors examined if a weighted bias for the expression level of the 8 genes might be useful to give a score rather than simply adding together expression levels?

Reviewer 2 Report

Kakodkar et al. investigate the role of aspartic aminopeptidase as biomarker in CLL. Although the topic is of some interest, the results as presented do not adequately support the conclusion. The design, description and statistics need substantial improvement.

There are many established prognostic factors in CLL, ZAP70 is only one of them. Authors need to perform multivariable analyses in well-characterized patient cohorts correcting for all known independent prognostic factors & risk classifications in CLL.

Authors need to be specific with respect to whether they claim that DNPEP expression is prognostic or predictive and proof this with adequately described datasets. For instance the finding that DNPEP expression correlates with Rai- or Binet stages is not sufficient to claim prognostic value, there could be simply an association. Why was the association with clinical parameters not performed in the 107 patients transcriptomic dataset?

The datasets used in the study are not clearly described and there are inconsistencies concerning the numbers of gene expression datasets used. Clinical characteristics are missing. Authors also need to provide references for the datasets e.g. for the transcriptomic data from the 107 CLL patients.

The text states that DNPEP was the strongest determinant of prognosis. This is based on a poorer separation when DNPEP was eliminated from the 8 gene panel. However, an analysis of DNPEP alone (including multivariable HR, with 95% CI, p-value and log-rank test) is missing to support this statement. Similarly, KM-plots and HRs for elimination of any of the other 8 genes are missing.

The statistics need to be described in more detail.

the experiments carried out with DI93293 are inadequate. How do authors exclude off-target effects? Statistics are missing. The additive effect with established treatments is minor.

Reviewer 3 Report

In the manuscript, the Authors investigate the expression of several kinases involved in BCR signaling, using available Gene Expression Profiling datasets to select a gene signature which correlated with BCR-signalling kinases overexpressed and/or overactivated in CLL. By this approach, Authors selected a number of genes, further tested for prognostic significance, identifying the DNPEP peptidase as a potential target.

The study presents several methodological flaws, many conclusions appear circumstantial and validation steps are lacking.

- The initial analysis is performed on 14 datasets downloaded from GEO and Arrayexpress, but no information is provided regarding the origin, characteristics or any reference whatsoever.

- ZAP70 was used as marker for aggressive CLL / enhanced BCR signaling: it is not known how ZAP-70 expression is investigated by the Authors in their CLL cohort and which cutoff has been employed (literature has described more than one method to investigate ZAP-70 expression which were not all concordant regarding the results in term of ZAP-70 positive cases); if they are using the GEP values, Authors have to consider that this approach has not been utilized to discriminate between ZAP70pos and ZAP70neg cases at least by large studies aimed at identifying the clinical imact of ZAP70 over-expression in CLL;  

- again, no information is provided regarding e.g. the IGHV mutational status, for which ZAP70 was often regarded as a surrogate, and/or the presence/absence of other major prognostic biomarkers (i.e. TP53 deletion and mutation, CD49d expression, 11q deletion etc) as well as of the more recently introduced mutations of genes like NOTCH1, BIRC3 or SF3B1; in this regard, both ZAP70 and CD38 expression have been unfortunately revised as biomarkers with less relevance than thought in the past (see eg. JCO, 32, 897, 2014)

- The relative HR for each of the 32 selected genes seems to be tested in univariate analysis (fig. 2B) but the top-scoring genes appear not to be the most critical contributors, as they cluster at the center of the waterfall plot in fig. 2D.

- How is tested the expression of these genes? GEP values? Over/below the median values?  Is it the same for ZAP70? What is the relative correlation between each of the 32 genes?

- With a very high correlation between these top-scoring genes, removing one would not change the overall HR; Authors should try to restrict the signature, removing highly correlated features and testing with a multivariate analysis. Actually, only 6 genes (plus ZAP70) have a significant impact in the prognosis of the 107 cases of the CLL cohort; it is hard to understand why all the 32 genes (and not only the 6 with a significant HR) have been utilized to build up the KM of fig 2C. 

- I do not understand how the HR of 37.79 is derived (in this calculation, which is the control cohort? Is it the one with all the 32 genes not expressed? This has to be better explained)

- The estimation of the DNPEP (and other genes) prognostic value was assessed using another Microarray dataset, but not validated in a second cohort. Also, no other prognostic factor is evaluated in a multivariate analysis, carried out by including the most relevant prognostic markers at the moment (eg IGHV gene status FISH staging and or CD49d expression and/or TP53 mutations) along with other variables like age (if the Authors investigate the OS as clinical readout) Rai/Binet staging at diagnosis/presentation, serum levels of beta2-microglobulin. The absence of these validations makes the validity of the suggested biomarker hardly acceptable.

- In-vitro experiments are conducted for very long treatment periods (up to 48h) and with very high concentrations of drugs (up to 10uM Idelalisib, Ibrutinib). Also, p values are not reported.

Reviewer 4 Report

The paper by Kakodkar et al. aims to identify new biomarkers of CLL aggressiveness and functionally link them to increased or re-wired BCR signaling. Through correlation analyses the Authors identidified a set of genes correlating with a known poor prognosis factor ZAP70 and other BCR-related kinases and developed a model to predict disease aggresiveness. Out of identified genes, the DNPEP gene has the strongest predictive value. Further, the Authors aimed to assess the pharmacologic inghibition of DNPEP enzyme in CLL cell line.

Although interesting, the paper needs some improvements and clarifications before being suitable for publication.

Major comments:

1) Is the new predictive gene set independent of currently known risk factors such as ZAP70 status, CD38 expression or IGHV mutation ? Was the multivariate analysis performed ?

2) Is it known if the patients characterized by the expression of the gene set related to increased BCR signaling components also better respond to ibrutinib or idelalisib in clinical trials ?

3) Figure 4A - there is no statistical test provided to support the claim that DNPEP inhibition significantly potentiates ABT-737 activity neither it can be seen from the figure (error bars overlapping). Please indicate which groups are compared with which test.

4) Figure 4D-E - the concentrations of Idelalisib and Ibrutinib (10uM both) are 10 and 100 fold higher than widely accepted as specific, respectively, and most likely show off target activity.

Minor comments:

1) Introduction, verses 77 and 78, the sentence ending with "activation of BcR kinases resulting in compensatory pro-survival signal transduction, such as induction of the apoptotic protein BCL2". First, it needs a reference. Second, as far as I am concerned the BCR signaling may induce expression of BCL-2 family members such as BFL-1 (Thijssen R et al. 2016) or MCL-1 (Bojarczuk et al. 2016), but not BCL-2 itself. The BCL-2 overexpression os believed to be caused by 13q14 deletion of miRNA 15/16 that negatively regulate BCL2.

2) How the authors propose that the new biomarker (DNPEP) expression is measured in primary samples? Is IHC feasible or the relative gene expression ? That should also be discussed